# Increased Mortality with Intermediate Ascitic Polymorphonuclear Cell Counts Amongst Patients with Cirrhosis: Time to Redefine the Care Approach

**DOI:** 10.3390/pathophysiology32040062

**Published:** 2025-11-11

**Authors:** Shahid Habib, Michael Ball, Chris Thomas, Traci Murakami, Nehali Patel, Sandeep Yarlagadda, Sarah Patel, Courtney Walker, Varun Takyar, Krunal Patel, Christian Domingues, Chiu-Hsieh Hsu

**Affiliations:** 1Liver Institute PLLC, Tucson, AZ 85716, USA; 2Transplant Department, HCA HealthONE Presbyterian St. Luke’s, Denver, CO 80218, USA; michael.ball@hcahealthone.com (M.B.); chris.thomas@hcahealthone.com (C.T.); 3Division of Gastroenterology, Department of Medicine, The Queen’s Medical Center, West Oahu, HI 96706, USA; traci.murakami@gmail.com; 4Southern California Permanente Medical Group, Division of Gastroenterology, Department of Medicine, Fontana, CA 92335, USA; humaandmsn@gmail.com; 5Department of Medicine, Fountain Hills Medical Center, Fountain Hills, AZ 85268, USA; syarlagadda@fhmcaz.com; 6Sonoran Sleep Center, Phoenix, AZ 85306, USA; huma840@gmail.com; 7Division of Gastroenterology, Department of Medicine, AdventHealth Porter Hospital, Denver, CO 80210, USA; courtney.walker.do@adventhealth.com; 8Division of Gastroenterology, Division of Medicine, Sutter Health, Antioch, CA 94513, USA; takyarv@sutterhealth.org; 9Borland Groover Specialists, Jacksonville, FL 32256, USA; hhabib@liverinstitutepllc.org; 10Division of Gastroenterology, Department of Medicine, Digestive Health Associates, Houston, TX 77024, USA; 11Mel and Enid Zuckerman College of Public Health, University of Arizona, Tucson, AZ 85724, USA; pchhsu@email.arizona.edu

**Keywords:** ascites, spontaneous bacterial peritonitis, cirrhosis, liver disease, polymorphonuclear cells, decompensation, peritonitis, ascites

## Abstract

Background: Spontaneous bacterial peritonitis (SBP) is a serious complication in patients with decompensated cirrhosis and ascites. Diagnosis typically relies on an ascitic polymorphonuclear (A-PMN) cell count ≥ 250 cells/high-power field (HPF). Methods: In this retrospective cohort study, 117 hospitalized patients with acute decompensation of chronic liver disease and a diagnostic paracentesis were evaluated. Clinical, laboratory, and imaging data were collected. Patients were stratified by A-PMN counts of ≤50, 51–249, or ≥250 cells/HPF. Additional analysis was performed with patients stratified by ascitic white blood cell (WBC) count and albumin. Mortality risk was assessed at 28, 90, and 365 days. Results: Patients with A-PMN ≤ 50 cells/HPF had the lowest 28-day mortality (8%). At 90 and 365 days, mortality risk was significantly higher for the A-PMN 51–249 cells/HPF group (90-day hazard ratio (HR) 3.55, *p* = 0.01; 365-day HR 2.43, *p* = 0.02), but not A-PMN ≥ 250 cells/HPF group (90-day HR 2.95, *p* = 0.1; 365-day HR 2.95, *p* = 0.2). Ascitic WBC count did not significantly predict mortality, though higher counts were associated with extraperitoneal infections. Ascitic fluid albumin ≤ 1.0 g/dL was independently associated with increased 365-day mortality (HR 3.53, *p* = 0.03). Conclusions: Binary SBP A-PMN thresholds may not adequately capture mortality risk in cirrhotic patients with ascites. Low ascitic albumin and intermediate A-PMN counts are associated with increased long-term mortality, suggesting the need for more nuanced diagnostic and prognostic criteria in SBP evaluation.

## 1. Introduction

As the 11th leading cause of mortality, chronic liver diseases and cirrhosis account for approximately 2 million annual deaths worldwide [1,2]. Cirrhosis significantly increases mortality risk—up to 10-fold—due to decompensating events such as ascites, variceal bleeding, hepatic encephalopathy, infections, and renal impairment [1]. Among these, ascites is the most common complication and carries a 15% mortality risk within one year and 44% within five years [3,4]. Spontaneous bacterial peritonitis (SBP) occurs when ascitic fluid becomes infected in the absence of a known primary source that can be surgically remedied [5,6,7]. Amongst cirrhotic patients, SBP is the most prevalent cause of bacterial infections with reported incidence rates of 11–14% among those hospitalized [8,9,10,11]. The presentation of SBP varies widely from asymptomatic to severe systemic decompensation requiring hospitalization. Mortality rates have been reported at 19.5% for 30-day in-hospital mortality following an initial SBP episode [12].

Amongst a large retrospective cohort of acute-on-chronic liver failure (ACLF) patients, SBP was associated with a 1.79-fold increase in 90-day mortality when compared to those without infection (*p* < 0.001; 95% confidence interval (CI): 1.58, 2.02) [13]. If not promptly managed, SBP mortality has been reported as high as 81.8% in severe cases [14]. Therefore, early recognition and timely treatment are critical to improving outcomes in SBP management and remains among the principle challenges in clinical management of this patient population. paramount to overall clinical success in SBP management are the principal challenges of early recognition and treatment.

A definitive diagnosis of SBP can be established by the isolation of a bacterial pathogen from ascitic fluid. However, the time required for culture growth and a high rate of negative cultures, up to 60% incidence, requires alternative diagnostic methods [15,16,17,18]. Analysis of ascitic fluid is therefore also used as an indicator with guidelines supporting the diagnosis of SBP with an ascitic polymorphonuclear (A-PMN) cell count of ≥250 cells/high-power field (HPF) [5,6,7]. Diagnostic models currently utilize absolute A-PMNs for this threshold, though relative A-PMN counts can also provide diagnostic value [19]. An emerging body of evidence suggests that relying on the binary classification of SBP vs. non-SBP based only on A-PMN counts may not adequately reflect mortality risk, particularly in patients with sub-diagnostic counts. We evaluated cirrhotic inpatients with diagnostic paracenteses and compared those above and below the diagnostic A-PMN threshold, looking to identify key factors associated with mortality risk.

## 2. Methods

The Institutional Review Board of the University of Arizona approved the study protocol, and the requirement for informed consent was waived due to the study’s retrospective nature. Appendix A describes the data extraction and handling process. International Classification of Disease, Ninth and Tenth Revision (ICD-9/10) codes were extracted for liver disease related diagnoses. Chart review was conducted with a predefined protocol to collect demographics, medical history, social history, and cirrhosis-related complications. Laboratory test results within 24 hours of admission were considered along with all available imaging to assess the diagnosis of cirrhosis or chronic liver disease, portal hypertension, air space disease, or stroke (if brain computed tomography (CT) performed). Primary liver disease was categorized as alcohol, viral, metabolic-associated, and other. The presence of acute precipitants was a major factor in acute deterioration of stable cirrhotic patients, regardless of prior decompensation. Four precipitants were defined: infections, drugs or toxins, alcohol consumption, and gastrointestinal (GI) bleeding. Infection was identified by positive cultures from any source, diagnostic laboratory work indicative of infection (e.g., A-PMN ≥ 250 cells/HPF, positive polymerase chain reaction (PCR) testing), and/or imaging supporting infection (e.g., chest X-ray evidencing pneumonia). Alcohol consumption was considered a precipitant if documentation indicated alcohol usage exceeding daily limit recommendations within the weeks prior to admission. Drugs or toxins were considered if use of a known hepatotoxin other than alcohol was documented within three months prior to admission. GI bleeding was defined by clinical evidence noted in chart documentation or procedural findings.

In addition to the above data, general aspects of each patient’s hospital course were also collected. These included infectious diagnoses, antimicrobial treatments, corticosteroids for alcohol hepatitis, in-hospital mortality, and total length of stay. Additional information regarding the use of medications, supplements, herbal products, and other toxins was also included. The living status of the entire final cohort was assessed and verified from the United States of America Social Security database.

Patients were eligible for inclusion if they were ≥18 years old, had chronic liver disease or cirrhosis (confirmed by history, biochemical markers, and/or imaging), and acute deterioration requiring hospitalization. Exclusion criteria were diagnosis of malignancy, incomplete clinical data, and evidence of cerebrovascular disease on head CT. Additionally, patients without ascites were excluded, as well as those with ascites who did not undergo diagnostic paracentesis. These criteria ensured only individuals with sufficient clinical documentation and confirmed ascitic fluid analysis were included, allowing for accurate evaluation of SBP and associated outcomes.

### Categorization of Cohort and Statistical Analysis

Eligible patients who met the criteria for acute decompensation requiring hospitalization were retrospectively evaluated. Disease severity was assessed using the Aspartate Aminotransferase to Platelet Ratio Index (APRI), Child-Turcotte Pugh (CTP) score, Fibrosis-4 (Fib-4) score, and Model for End-Stage Liver Disease (MELD) score. Clinical features are presented as mean ± standard deviation (SD) for continuous variables and frequency and percentage for categorical variables. Fisher’s exact test was used to compare categorical variables between groups.

Mortality risk based upon ascites fluid analysis was assessed related to A-PMN count, white blood cell (WBC) count, and albumin levels. For A-PMN analysis, patients were categorized into three groups based on cell count: ≤50, 51–249, and ≥250 cells/HPF. Ascitic WBC count was initially grouped into three categories: ≤100, 101–500, and >500 cells/HPF, and then just dichotomized as either above or below 100 cells/HPF. Lower cutoffs were determined upon the lowest observed mortality risk. Albumin was analyzed as both a continuous and dichotomized variable using a cutoff of 1 g/dL.

All historic characteristics, clinical features upon admission, and clinic courses were evaluated. All groups were analyzed for infection rates and survival outcomes using descriptive statistical analysis to calculate percentages, mean, and ratio; with Pearson Correlation Coefficient to assess the correlation and *p*-value. Further, Cox regression was used to identify factors linked with survival and evaluate the predictability for each fitted model, in which C-statistic and receiver operating characteristic (ROC) curves were derived. Cox regression model was used to evaluate the mortality at 28, 90, and 365 days. All statistical analyses were performed in SPSS Statistics, version 30 (IBM Corp., Armonk, NY, USA).

## 3. Results

### 3.1. Baseline Characteristics of the Cohort

Of the 1865 patients initially screened, 521 (27.9%) met the predefined inclusion and exclusion criteria. Sixty-four patients were excluded due to incomplete data, leaving 457 eligible patients. Among these, 205 (44.9%) presented with clinical or radiologic evidence of ascites. Of the patients with ascites, 118 (57.6%) underwent diagnostic paracentesis during hospitalization. After excluding one patient’s repeat encounter, 117 unique patients were included in the final analysis.

Basic fungal workup, such as blood or ascites fluid, cultures, and potassium hydroxide staining, was not performed in most patients. Among the 117 patients included, A-PMN counts were ≤50 cells/HPF for 86 patients (73.5%), 51–249 cells/HPF for 18 patients (15.4%), and ≥250 cells/HPF for 13 patients (11.1%). Most baseline characteristics were overall similar between groups (Table 1). However, statistically significant differences were found for antibiotic indication use and infection status. The A-PMN ≥ 250 cells/HPF group had a higher proportion of patients on antibiotics for treatment and any infection. This finding was expected based on the stratification of patients by A-PMN with the ≥250 cells/HPF group representing those with SBP and treatment antibiotics. Most patients were male (77.8%) and mean age was 54.9 ± 10 years. The most common etiologies of liver disease were alcohol-associated cirrhosis and hepatitis C (HCV), each accounting for 43 patients (37.7%). Diabetes mellitus was present in 31 patients (26.5%) and chronic kidney disease in 20 patients (17.1%). Alcohol use within 6 months prior to admission was found in more than half of cohort (54%). Most patients did not meet criteria for ACLF defined by the European Association for the Study of the Liver (ACLF-EASL) More importantly, the severity of liver disease did not differ significantly between groups. Numerically, MELD scores rose for patients in a higher A-PMN cell count group. Mean MELD scores were 19.3 ± 8.5, 20 ± 8.7, and 22.3 ± 11.3 for the A-PMN ≤ 50, 51–249, and ≥250 cells/HPF groups, respectively. These differences were not statistically significant, nor clinically significant with a mean difference of 3 MELD score points from the A-PMN ≤ 50 to ≥250 cell/HPF groups. The overall means for APRI, CTP score, Fib-4 score, and MELD score were 2.6 ± 4.7, 12.1 ± 11.2, 8.2 ± 10.9, and 19.7 ± 8.8, respectively. These scores were in alignment with the severity of disease expected for this patient sample.

### 3.2. Mortality Risk Assessment

In an adjusted analysis for covariables age, gender, DM, MELD, infection diagnosis, and alcohol use, the lowest risk of mortality was at 28 days (8%) in the A-PMN ≤ 50 cells/HPF, which was similar to the WBC count ≤ 100 cells/HPF. No statistically significant differences were observed in 28-day mortality between A-PMN ≤ 50 cells/HPF group and 51–249 cells/HPF group (HR 1.64; *p* = 0.43; 95% CI: 0.49, 5.51) or ≥250 cells/HPF group (HR 1.77; *p* = 0.44; 95% CI: 0.41, 7.6) (Figure 1). The 90-day adjusted analysis found a significantly increased mortality risk between the A-PMN ≤ 50 cell/HPF group and 51–249 cells/HPF group (HR 3.55; *p* = 0.01; 95% CI: 1.36, 9.25). The ≥250 cells/HPF group had numerically, but not significantly, higher mortality when compared to the ≤50 cells/HPF group (HR 2.95; *p* = 0.1; 95% CI: 0.82, 10.64) (Figure 2). At 365 days, the A-PMN 51–249 cells/HPF continued to show significantly increase mortality compared to the A-PMN ≤ 50 cell/HPF (HR 2.43; *p* = 0.02; 95% CI: 1.12, 5.23). Again, the mortality for the ≥250 cells/HPF group was numerically higher than the A-PMN ≤ 50 cells/HPF groups, but not significantly so (HR 2.95; *p* = 0.2; 95% CI: 0.71, 4.99) (Figure 3).

Analysis based on total ascites fluid WBC count failed to show any statistically significant differences in mortality risk at 28, 90, and 365 days from index admission. Nonetheless, mortality risk rose with time among patients with WBC count > 100 cells/HPF from 28, 90, and 365 days (HR 0.95, 1.43, and 1.49, respectively). Interestingly, all patients with ascites fluid WBC count > 500 cells/HPF and 57% of those with ascites fluid WBC count of 101–500 cells/HPF had been diagnosed with extraperitoneal infection. There was no correlation between ascitic fluid cell counts and extra-peritoneal infection, as most patients had low cell counts in ascites fluid.

Analysis of ascitic albumin also revealed correlation with mortality risk. Mean ascitic albumin for the cohort was 0.99 ± 0.51 g/dL. Analysis was performed with ascitic albumin as a continuous and dichotomous variable. On bivariate analysis of patient survival time using ascitic albumin as a continuous variable, there was a statistically significant correlation between lower ascitic albumin and mortality (r = 0.2, *p* = 0.05, 95% CI: 0.0, 0.39). As a dichotomized variable, patients with ascites albumin ≤ 1.0 g/dL were associated with higher mortality compared to patients with ascites fluid albumin > 1.0 g/dL at any time. Again, the risk increased with time and became statistically significant only at 365 days from index admission (HR 3.53, *p* = 0.03; 95% CI: 1.07, 11.67). The mortality rates at 28, 90, and 365 days of index admission were 8%, 16%, and 51%, respectively, whereas the mortality rate at 28, 90 and 365 days of index admission in patients with higher ascites fluid albumin are 4%, 5%, and 12%, respectively.

## 4. Discussion

In this retrospective cohort study of hospitalized patients with cirrhosis and acute decompensation, we found that patients with A-PMN counts of 51–249 cells/HPF—below the traditional diagnostic threshold for SBP—had significantly increased long-term mortality compared to those with A-PMN counts ≤ 50 cells/HPF. Furthermore, low ascitic fluid albumin (≤1.0 g/dL) was independently associated with increased 365-day mortality, underscoring its potential role as a prognostic marker.

Although ascites is a key component of the CTP score, it is not included in the MELD 3.0 or EASL-Chronic Liver Failure ACLF prognostic models. The CTP score is no longer routinely used in clinical practice, yet its individual components—aside from ascites—are incorporated into modern prognostic tools. Our findings demonstrate that ascitic fluid analysis remains an important prognostic indicator, independently associated with both short- and long-term mortality, regardless of MELD score or ACLF status.

The lowest observed 28-day mortality was 8%, which corresponded to patients with A-PMN ≤ 50 cells/HPF or WBC counts ≤ 100 cells/HPF. These values may serve as reference points for “normal” cell counts in the context of decompensated cirrhosis. Additionally, compared to patients with A-PMN ≤ 50 cells/HPF, those with sub-diagnostic A-PMN counts (51–249 cells/HPF) and those meeting the SBP threshold (≥250 cells/HPF) exhibited higher mortality at 28, 90, and 365 days, though this only reached statistical significance for those with A-PMN 51–249 cell/HPF at 90 and 365 days. When compared to the A-PMN 51–249 cells/HPF group, the trend towards decreased mortality amongst the A-PMN ≥ 250 cells/HPF over time is suggestive of mortality mitigation through antibiotic treatment. While our study was not designed to evaluate the impact of antibiotic therapy, this remains an area for future study; particularly for the role antibiotic therapy may have on mortality rates with A-PMN of 51–249 cells/HPF. Prophylactic or treatment antibiotics may have a key role in these sub-diagnostic A-PMN patients. Interestingly, ascitic fluid WBC count did not correlate with mortality, and bivariate analysis revealed no association between ascitic fluid cell counts and extra-hepatic infections.

Current guidelines define SBP using an A-PMN threshold of ≥250 cells/HPF [5,6,7]. While appropriate for diagnosis, this threshold may not adequately capture other high-risk subgroups amongst cirrhotic patients with ascites. Our findings add to the literature by identifying higher mortality rates for cirrhotic patients with intermediate A-PMN counts. Importantly, the increased mortality observed in this subgroup was independent of MELD score and ACLF status. These patients may have subclinical infections or an elevated inflammatory response, setting them apart as a unique population that may benefit from closer monitoring or earlier intervention. Both our data and the growing body of literature around sub-diagnostic A-PMN counts lends credence to this notion.

Sub-diagnostic A-PMN counts ≥ 100 cells/HPF, coupled with CTP stage C and serum sodium < 125 mmol/L indicate a significant risk for developing SBP amongst SBP naïve patients [20]. Relative A-PMN counts, determined as absolute A-PMN count divided by the total ascitic leukocyte count, offer an alternative method for future SBP prediction [19]. While our data were not analyzed for subsequent encounters and SBP development, this is a potential future direction for investigation.

Comparable 1-year mortality risk has been identified for those with an A-PMN count of 125–250 cells/HPF and >250 cells/HPF at 75% and 80.9%, respectively, versus 40.5% with A-PMN counts < 125 cells/HPF (*p* = 0.016). In the same study, an increase of 100 cells/HPF in A-PMN counts from one paracentesis to another was also indicative of increased mortality [21]. From a different perspective, patients with A-PMN counts < 250 cells/HPF had a 10% increased risk of death with every 25-unit increase in A-PMN percentage [22]. We reported findings of a single paracentesis and the outcomes thereafter while serial paracentesis results may further define and delineate risk periods [23]. However, clinicians will often find themselves faced with results of a solitary paracentesis and need to know the risk that lies therein.

Aside from A-PMN, several alternative markers in ascitic fluid have been evaluated to categorize risk. However, these may be unavailable or untimely compared to A-PMN counts, which exist within the standard of care when evaluating ascitic fluid [24,25,26,27]. Primarily due to the retrospective design, our study did not evaluate all potential markers. However, the use of standard ascitic analysis provides current generalizability to what clinicians encounter in practice. Within our analysis, the significance of ascitic albumin concentrations was of particular importance. In this study, the association between low ascitic albumin (≤1.0 g/dL) and higher 365-day mortality further supports the need to reconsider current risk thresholds. Taken together, our findings suggest that both intermediate A-PMN counts (51–249 cells/HPF) and low ascitic albumin may warrant consideration for primary antibiotic prophylaxis. These results align with emerging evidence that SBP may exist on a spectrum, and that rigid diagnostic cutoffs can fail to identify some patients at risk for poor outcomes.

Our study has several limitations. The retrospective nature allows for the inherently uncontrolled variance of clinical practice. We were unable to describe all courses of antimicrobial therapy but rather defined them based on intention and duration. While not ideal, the influence of this would be expected to be seen earlier with mortality rates at 28 days, for which we did not observe a difference from the A-PMN ≤ 50 cells/HPF and the 51–249 cells/HPF or ≥250 cells/HPF groups. We were unable to provide the cause of death due to the retrospective nature of the study design and the multi-factorial nature of mortality amongst cirrhotic patients. Patients included in this study also represent a relatively high-risk population as evidenced by the average MELD scores ranging from 19 to 22. We submit that this level of mortality risk represents real-world clinical practice for decompensated cirrhotic inpatients with ascites and concern for SBP. Lastly, our sample size for A-PMN counts of 51–249 and ≥250 cells/HPF were small. Expansion of these subgroups may have allowed for further conclusions to be drawn. The small number of patients in these groups also limited the ability to perform a logistic regression for mortality stratified by A-PMN counts. Furthermore, the analysis for mortality risk may be underpowered based on the sample sizes in the A-PMN ≥ 250 cells/HPF. While we found no statistically significant differences in mortality and these patients carry a high-risk of death, a larger sample size may have elucidated differences our analysis was not powered to detect and could have bolstered our findings. However, this was subject to the retrospective nature of the study design. Further analysis comprised of increased sample sizes is warranted to confirm the findings we present here.

Patients with A-PMN counts nearing the 250 cells/HPF threshold carry a high-risk of mortality, comparable to those with confirmed SBP. As SBP patients are indicated for antibiotic treatment and prophylaxis, the question then becomes what the optimum threshold for antibiotic initiation is to lower mortality in patients without diagnostic SBP A-PMN counts. Furthermore, is this assessment best understood as A-PMN absolute counts, relative counts, percentages, or through composite evaluation of non-PMN markers in conjunction? An idea that is already gaining traction is the use of artificial intelligence and machine learning to open future diagnostic pathways, namely ones that utilize noninvasive samples [28]. This approach could also further mitigate the risk faced by these patients. A prospective, randomized controlled trial would help delineate the true role of antibiotic prophylaxis in this sub-diagnostic A-PMN population.

## 5. Conclusions

Patients with sub-diagnostic A-PMN counts of 51–249 cells/HPF carry a high-risk of 90- and 365-day mortality and warrant consideration for further assessment and mitigation of this risk. Current practices utilizing a binary A-PMN SBP diagnostic model may fail to adequately address these patients by categorizing them as non-SBP.

## Figures and Tables

**Figure 1 pathophysiology-32-00062-f001:**
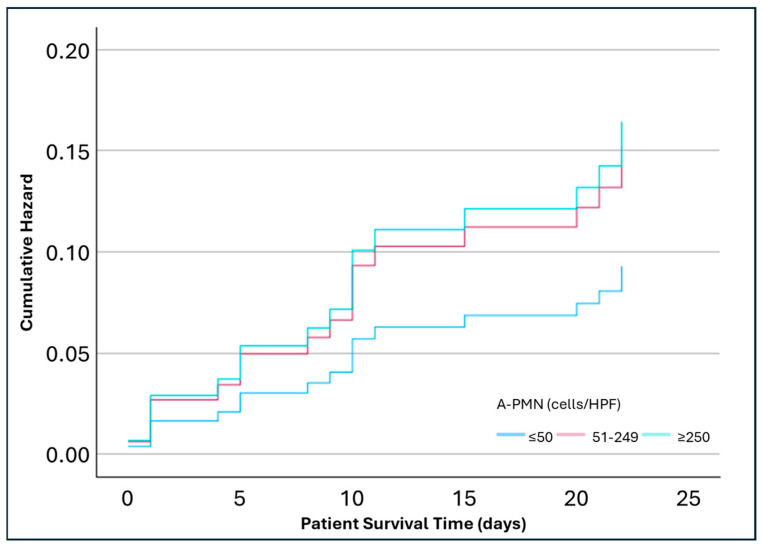
Adjusted analysis of mortality at 28 days based on ascitic polymorphonuclear cell counts (n = 117). Analysis adjusted for covariables age, gender, diabetes mellitus, Model for End-Stage Liver Disease score, infection status, and alcohol use. Hazard Ratio (A-PMN 51–249 cells/HPF vs. ≤50 cells/HPF) = 1.64 (*p* = 0.43; 95% CI: 0.49, 5.51). Hazard Ratio (A-PMN ≥ 250 cells/HPF vs. ≤50 cells/HPF) = 1.77 (*p* = 0.44; 95% CI: 0.41, 7.6).

**Figure 2 pathophysiology-32-00062-f002:**
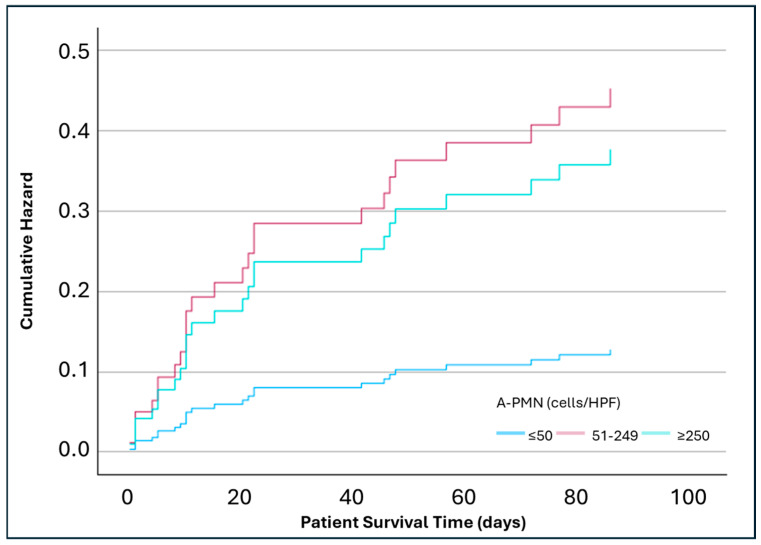
Adjusted analysis of mortality at 90 days based on ascitic polymorphonuclear cell counts (n = 117). Analysis adjusted for covariables age, gender, diabetes mellitus, Model for End-Stage Liver Disease score, infection status, and alcohol use. Hazard Ratio (A-PMN 51–249 cells/HPF vs. ≤50 cells/HPF) = 3.55 (*p* = 0.01; 95% CI: 1.36, 9.25). Hazard Ratio (A-PMN ≥ 250 cells/HPF vs. ≤50 cells/HPF) = 2.95 (*p* = 0.1; 95% CI: 0.82, 10.64).

**Figure 3 pathophysiology-32-00062-f003:**
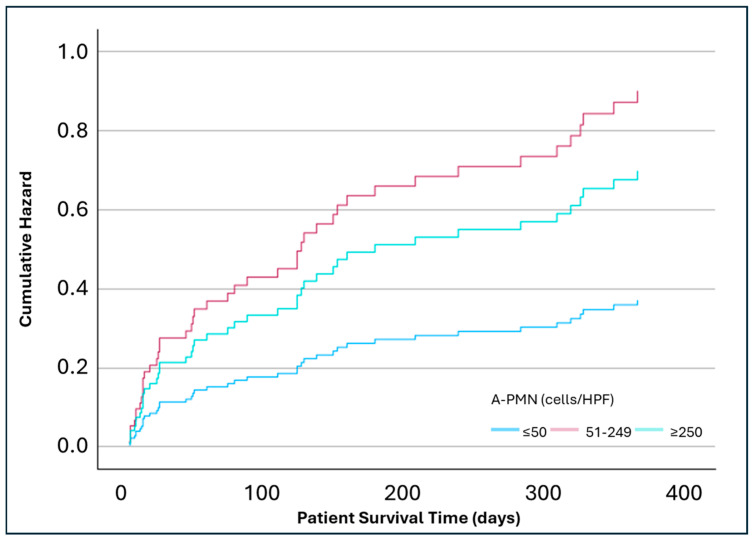
Adjusted analysis of mortality at 365 days based on ascitic polymorphonuclear cell counts (n = 117). Analysis adjusted for covariables age, gender, diabetes mellitus, Model for End-Stage Liver Disease score, infection status, and alcohol use. Hazard Ratio (A-PMN 51–249 cells/HPF vs. ≤50 cells/HPF) = 2.43 (*p* = 0.02; 95% CI: 1.12, 5.23). Hazard Ratio (A-PMN ≥ 250 cells/HPF vs. ≤50 cells/HPF) = 2.95 (*p* = 0.2; 95% CI: 0.71, 4.99).

**Table 1 pathophysiology-32-00062-t001:** Baseline characteristics.

Characteristic	A-PMN ≤50 Cells/HPF (n = 86)	A-PMN 51–249 Cells/HPF (n = 18)	A-PMN ≥250 Cells/HPF (n = 13)	Total (n = 117)	*p* Value
Male gender—no. (%)	66 (76.7)	13 (72.2)	12 (92.3)	91 (77.8)	0.375 ^A^
Mean age (SD)—years	54.3 (10.7)	56.7 (7.2)	56.7 (8.2)	54.9 (10)	0.514 ^B^
Cirrhosis etiology—no./total no. (%)					0.54 ^A^
Alcohol	33/84 (39.3)	5/17 (29.4)	5/13 (38.5)	43/114 (37.7)	
Hepatitis C virus	28/84 (33.3)	9/17 (52.9)	6/13 (46.2)	43/114 (37.7)	
Hepatitis B virus	4/84 (4.8)	0/17 (0)	1/13 (7.7)	5/114 (4.4)	
Metabolic-associated liver disease	10/84 (11.9)	0/17 (0)	0/13 (0)	10/114 (8.8)	
Autoimmune hepatitis	1/84 (1.2)	1/17 (5.9)	0/13 (0)	2/114 (1.8)	
Unknown/Cryptogenic	8/84 (9.5)	2/17 (11.8)	1/13 (7.7)	11/114 (9.6)	
Diabetes mellitus—no. (%)	26 (30.2)	1 (5.6)	4 (30.8)	31 (26.5)	0.091 ^A^
Chronic kidney disease—no. (%)	12 (14)	5 (27.8)	3 (23.1)	20 (17.1)	0.305 ^A^
Alcohol use—no./no. total (%)					0.969 ^A^
No	26/83 (31.3)	7/18 (38.9)	4/12 (33.3)	37/113 (32.7)	
Yes, <6 months prior	46/83 (55.4)	9/18 (50)	6/12 (50)	61/113 (54)	
Yes, >6 months prior	11/83 (13.3)	2/18 (11.1)	2/12 (16.7)	15/113 (13.3)	
ACLF-EASL level 1 minimum criteria met ^C^—no./no. total (%)	29/86 (33.7)	8/18 (44.4)	7/13 (53.8)	44/117 (37.6)	0.305 ^A^
APRI (n = 117)—mean (SD)	2.4 (2.8)	4.3 (10.2)	1.4 (1.5)	2.6 (4.7)	0.195 ^B^
CTP score (n = 117)—mean (SD)	12.1 (1.3)	12.1 (1.3)	11.8 (0.6)	12.1 (11.2)	0.612 ^B^
Fib-4 score (n = 117)—mean (SD)	7.7 (6.7)	12.5 (23.3)	5.5 (5.3)	8.2 (10.9)	0.149 ^B^
MELD score (n = 115)—mean (SD)	19.3 (8.5)	20.0 (8.7)	22.3 (11.3)	19.7 (8.8)	0.510 ^B^
Antibiotic agents—no./no. total (%)					0.648 ^A^
1 agent	45/69 (65.2)	7/15 (46.7)	7/12 (58.3)	59/96 (61.5)	
≥2 agents	23/69 (33.3)	8/15 (53.3)	5/12 (41.7)	36/96 (37.5)	
Antibiotic initiation time from admission—no./no. total (%)					0.850 ^A^
<24 h	62/65 (95.4)	14/15 (93.3)	11/12 (91.7)	87/92 (94.6)	
>24 h	3/65 (4.6)	1/15 (6.7)	1/12 (8.3)	5/92 (5.4)	
Reason for antibiotic use (for any infection—no./no. total (%)					0.015 ^A^
Prophylaxis	30/68 (44.1)	5/15 (33.3)	2/12 (16.7)	37/95 (38.9)	
Preemptive	17/68 (25)	4/15 (26.7)	0/12 (0)	21/95 (22.1)	
Treatment	21/68 (30.9)	6/15 (40)	10/12 (83.3)	37/95 (38.9)	
Infection status (any infection)—no. (%)	52 (60.5)	11 (61.1)	13 (100)	76 (65)	0.019 ^A^

^A^: Pearson Chi-Square; ^B^: ANOVA; ^C^: One organ failure and serum creatinine 1.5–1.9 mg/dL. ACLF-EASL, Acute-on-Chronic Liver Failure-European Association for the Study of the Liver; APRI, Aspartate aminotransferase to Platelet Ratio Index; CTP, Child Turcotte Pugh; Fib-4, Fibrosis 4; MELD, Model for End-Stage Liver Disease; SD, standard deviation.

## Data Availability

The data supporting the findings of this study are available from the corresponding author upon reasonable request.

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
