# Peer review of "Increased Mortality with Intermediate Ascitic Polymorphonuclear Cell Counts Amongst Patients with Cirrhosis: Time to Redefine the Care Approach"

_pathophysiology, 2025, doi:10.3390/pathophysiology32040062_

Round 1

Reviewer 1 Report

Comments and Suggestions for Authors

Thank you for this opportunity to review this retrospective cohort study, which investigates the prognostic significance of ascitic polymorphonuclear cell counts in hospitalized patients with decompensated cirrhosis. Based on a sub-cohort of 117 patients who underwent paracentesis, the study concludes that patients with intermediate A-PMN counts have a long-term mortality risk that is not significantly different from patients diagnosed with SBP, while both groups have a higher risk than patients with low A-PMN counts. The study addresses what could be an interesting point in the management of cirrhotic ascites. However, the manuscript has methodological and statistical limitations that substantially weaken the confidence in its conclusions. I strongly recommend that the authors follow-up on the reported suggestions, which are further divided in major and minor areas for improvement, before they eventually resubmit.

Major Areas for Improvement

  1. The study's main conclusion rests on the finding of no significant difference in mortality between the intermediate A-PMN group (n=18) and the SBP (A-PMN ≥250) group (n=13). With such small sample sizes, to me the analysis is severely underpowered. The absence of a statistically significant difference in this context cannot be reliably interpreted as evidence of equivalent risk; rather, it may reflect an inability to detect a true difference if one exists. This limitation directly impacts the central assertion of the paper and should be acknowledged as a potential significant limitation and/or with further analyses before the authors resubmit.
  2. The manuscript structure is confusing due to the use of a “full cohort” (n=457) and a “sub-cohort” (n=117). The stated exclusion criteria note that patients who did not undergo diagnostic paracentesis were excluded. However, the initial analyses and figures are based on the full cohort, which explicitly includes a large group of patients with ascites who did not have a paracentesis (n=85). The authors need to clarify this, as it creates a central contradiction in the description of the study population and makes the overall analytical strategy significantly difficult to follow.
  3. The mortality analysis presented in Figure 2, which compares five distinct clinical groups from the full cohort is an unadjusted analysis. Presenting unadjusted survival curves for such heterogeneous groups is highly susceptible to confounding and provides limited meaningful insight into the independent mortality risk associated with each condition. Authors need to perform a logistic regression, taking into account all potentially confounding variables in the relationship they analyze. Furthermore, they need to revisit the entire statistical approach to the manuscript.
  4. The text describes the baseline characteristics of the three A-PMN sub-cohort groups as "overall similar". However, Table 1 shows clinically relevant differences in mean MELD score (19.3, 20.0, and 22.3 for the low, intermediate, and high A-PMN groups, respectively) and a statistically significant difference in the reasons for antibiotic administration. The assertion of baseline similarity is not fully supported by the data presented and authors are kindly asked to better specify or find it as a new limitation to the study.

Minor Areas for Improvement

  1. The manuscript uses the units "cells/HPF" (high power field) and "cells/µL" interchangeably when discussing A-PMN counts. While these are often used as surrogates in clinical practice, a consistent unit should be used throughout the manuscript for clarity and precision. Please correct.
  2. The legends for Figure 1 and Figure 3 state that the analyses are adjusted and list the co-variables. However, the tables embedded within the figures, which contain the hazard ratios and p-values, do not explicitly state whether these specific results are from the adjusted or an unadjusted model, so to say the least it is difficult to interpretate such results.
Comments on the Quality of English Language

An overall revision for spelling and grammar errors is required

Reviewer 2 Report

Comments and Suggestions for Authors

Authors evaluated ascitic PMN or WBC in patients with chronic liver diseases. As the results, authors found intermediate ascitic PMN counts are associated with increased mortality. This study was interesting, but several issues remained to be addressed.

  1. In patients with intermediate OMN counts, increased liver fibrosis was found. It should be considered.
  2. The definition of SBP in preset study should be clarified. In this study, 17 patients with SBP were included. But, only 13 patients with A-PMN>250 were found.
  3. The cause of mortality should be clarified. 
  4. The degree of portal hypertension and its contribution to the pathogenesis should be considered.
  5. In KM analyses, KM curve colors in 1, 3 and 5 are hardly to be distinguished. Different colors are desirable. 

Round 2

Reviewer 1 Report

Comments and Suggestions for Authors

I personally thank the authors for addressing the required amendments. The manuscript has largely improved. 

Reviewer 2 Report

Comments and Suggestions for Authors

Revised manuscript was well-addressed to the reviewer's comments and well-written. It has been improved in its clinical significance.